# LLM Circuit Analyses Are Consistent Across Training and Scale

**Curt Tigges**
EleutherAI
curt@eleuther.ai

**Michael Hanna**
ILLC, University of Amsterdam
m.w.hanna@uva.nl

**Qinan Yu**
Brown University
qinan_yu@brown.edu

**Stella Biderman**
EleutherAI
stella@eleuther.ai

## Abstract

Most currently deployed LLMs undergo continuous training or additional finetuning. By contrast, most research into LLMs' internal mechanisms focuses on models at one snapshot in time (the end of pre-training), raising the question of whether their results generalize to real-world settings. Existing studies of mechanisms over time focus on encoder-only or toy models, which differ significantly from most deployed models. In this study, we track how model mechanisms, operationalized as circuits, emerge and evolve across 300 billion tokens of training in decoder-only LLMs, in models ranging from 70 million to 2.8 billion parameters. We find that task abilities and the functional components that support them emerge consistently at similar token counts across scale. Moreover, although such components may be implemented by different attention heads over time, the overarching algorithm that they implement remains. Surprisingly, both these algorithms and the types of components involved therein tend to replicate across model scale. Finally, we find that circuit size correlates with model size and can fluctuate considerably over time even when the same algorithm is implemented. These results suggest that circuit analyses conducted on small models at the end of pre-training can provide insights that still apply after additional training and over model scale.

## 1 Introduction

As LLMs' capabilities have grown, so has interest in characterizing their mechanisms. Recent work in mechanistic interpretability often seeks to do so via circuits: computational subgraphs that explain task-solving mechanisms [71, 32, 48, 40]. Circuits can be found and verified using a variety of methods, [20, 65, 33, 38, 23] with the aim of reverse-engineering models' task-solving algorithms.

Though much circuits research is motivated by LLMs' capabilities, the setting in which such research is performed often differs from that of currently deployed models. Crucially, while most LLM circuits work [71, 32, 48, 40, 67] studies models at the end of pre-training, currently deployed models often undergo continuous training [55, 2, 66] or are fine-tuned for specific tasks [17, 34]. More generally, models are trained with varying amounts of data and circuits can be sampled at any checkpoint; to what extent do those circuits hold as training continues? Other subfields of interpretability have studied model development during training [35, 13, 72, 16, 12], but similar work on LLM mechanisms is scarce. Existing mechanistic work over training has studied syntactic attention structures and induction heads [54, 14, 63], but has focused on small encoder or toy models. Prakash et al. [56] examines circuits in 7-billion-parameter models post-finetuning, but the evolution of circuits during

38th Conference on Neural Information Processing Systems (NeurIPS 2024).

pre-training remains unexplored. This raises questions about whether circuit analyses will generalize if the model in question is further trained on a wide distribution of data.

We address this issue by exploring when and how circuits and their components emerge during training, and their consistency across training and different model scales.[1] We study circuits in models from the Pythia suite [5] across 300 billion tokens, at scales from 70 million to 2.8 billion parameters. We supplement this with additional data from models ranging up to 12 billion parameters. Our results suggest remarkable consistency in circuits and their attributes across scale and training. We summarize our contributions as follows:

**Performance acquisition and functional component emergence are similar across scale:** Task ability acquisition rates tend to reach a maximum at similar token counts across different model sizes. Functional components like name mover heads, copy suppression heads, and successor heads also emerge consistently at similar points across scales, paralleling previous findings that induction heads emerge at roughly 2B-5B tokens across models of all scales [54].

**Circuit algorithms often remain stable despite component-level fluctuations:** Analysis of the IOI circuit across training and scale reveals that even when individual components change, the overall algorithm remains consistent, indicating a certain level of algorithmic stability. In addition to stability across time, we find that the algorithm also tends to be similar for dramatically different model scales, suggesting that some currently-identified circuits may generalize.

**Graph-level circuit attributes vary across training but correlate with model size:** Once primary functionalities emerge, the circuit subgraph constituents tend to stabilize (with circuits in larger models showing higher stability), but we also observe exceptions: constituents can shift significantly even late in training. Circuits in larger models require more components, with circuit sizes positively correlating with model scale.

**Taken as a whole, our results suggest that circuit analysis generalizes well** over both training and scale even in the face of component and circuit size changes, and that circuits studied at the end of training in smaller models can indeed be informative for larger models as well as for models with longer training runs. We hope to see this validated for other circuits, especially more complex ones and those based on SAE latents, confirming our initial findings.

## 2 Methods

### 2.1 Circuits

A **circuit** [53, 22, 71] is the minimal computational subgraph of a model that is faithful to its behavior on a given task. At a high level, this means that circuits describe the components of a model that the model uses to perform the task; in this paper, the components that we study, and thus the nodes in our circuit, are the model's attention heads and multi-layer perceptrons (MLPs). A task, within the circuits framework, is defined by inputs, expected outputs, and a (continuous) metric that measures model performance on the task. For example, in the indirect object identification (IOI, [71]) task, the LM receives inputs like "When John and Mary went to the store, John gave a drink to", and is expected to output *Mary*, rather than *John*. We can measure the extent to which the LM fulfills our expectations by measuring the difference in logits assigned to *Mary* and *John*.

Circuits are useful objects of study because we can verify that they are *faithful* to LM behavior on the given task. We say that a circuit is faithful if we can corrupt all nodes and edges outside the circuit without changing model behavior on the task. Concretely, we test faithfulness by running the model on normal input, while replacing the activations corresponding to edges outside our circuit, with activations from a corrupted input, which elicits very different model behavior. In the above case, our corrupted input could instead be "When John and Mary went to the store, Mary gave a drink to", eliciting *John* over *Mary*. If the circuit still predicts *Mary*, rather than *John*, it is faithful. As circuits are often small, including less than 5% of model edges, this faithfulness test corrupts most of the model, thus guaranteeing that circuits capture a small set of task-relevant model mechanisms. For more details on the circuits framework, see prior work and surveys [20, 33, 24].

---

[1]Though we do not study fine-tuning or other post-training techniques, observing the types of changes that can occur over the course of pre-training may shed light on the changes that can occur in circuits more generally.

Circuits have a number of advantages over other interpretability frameworks. As computational subgraphs of the model that flow from its inputs to its outputs, they provide complete explanations for a model's mechanisms. Moreover, their faithfulness, verified using a causal test, makes them more reliable explanations. This stands in contrast to probing [3], which only offers layer-representation-level explanations, and can be unfaithful, capturing features unused by the model [21]. Similarly, input attributions [62, 64] only address which input tokens are used, and may be unreliable [1, 7].

## 2.2 Circuit Finding

In order to find faithful circuits at scale over many checkpoints, we use efficient, attribution-based circuit finding methods. Such methods score the importance of all edges in a model's graph in a fixed number of forward and backward passes, independent of model size; though other patching-based circuit-finding methods [20] are more accurate, they are too slow, requiring a number of forward passes that grows with model size. From the many existing attribution methods [50, 23, 38], we select edge attribution patching with integrated gradients (EAP-IG [33]) due to its faithful circuit-finding ability. Much like its predecessor, edge attribution patching (EAP [50]), EAP-IG assigns each edge an importance score using a gradient-based approximation of the change in loss that would occur if that edge were corrupted; however, EAP-IG yields more faithful circuits with fewer edges.

After running EAP-IG to score each edge, we define our circuit by greedily searching for the edges with the highest absolute score. We search for the minimal circuit that achieves at least 80% of the whole model's performance on the task. We do this using binary search over circuit sizes; the initial search space ranges from 1 edge to 5% of the model's edges. The high faithfulness threshold we set gives us confidence that our circuits capture most model mechanisms used on the given task. However, ensuring that a circuit is entirely complete, containing all relevant model nodes and edges, is challenging, and no definitive method of verifying this has emerged. The most notable existing method, from Wang et al. [71], requires comparing circuit and model performance under a wide variety of ablations, and is seldom used due to its complexity and computational cost.

## 2.3 Models

We study Biderman et al.'s [5] Pythia model suite, a collection of open-source autoregressive language models that includes intermediate training checkpoints. Though we could train our own language models or use another model suite with intermediate checkpoints [61, 42, 30], Pythia is unique in providing checkpoints for models at a variety of scales and training configurations.[2] Each model in the Pythia suite has 154 checkpoints: 11 of these correspond to the model after 0, 1, 2, 4, ..., and 512 steps of training; the remaining 143 correspond to 1000, 2000, ..., and 143,000 steps. We find circuits at each of these checkpoints. As Pythia uses a uniform batch size of 2.1 million tokens, these models are trained on far more tokens (300 billion) than those in existing studies of model internals over time. We study models of varying sizes, from 70 million to 12 billion parameters.

## 2.4 Tasks

We analyze the mechanisms behind four different tasks taken from the (mechanistic) interpretability literature. We choose these tasks because they are simple and feasible for even the smaller models we study. Moreover, as existing work has already studied them in other models, we have clues as to how our models likely perform these tasks; to verify that our models use similar circuits we briefly analyze our models' indirect object identification and greater-than circuits in Appendix A. The other task are MLP-dominant and do not involve much attention head activity; for these circuits, we verify that this is still the case in Pythia models.

**Indirect Object Identification**   The indirect object identification (IOI, [71]) task feeds models inputs such as "When John and Mary went to the store, John gave a drink to"; models should prefer *Mary* over *John*. Corrupted inputs, like "When John and Mary went to the store, Mary gave a drink to", reverse model preferences. We measure model behavior via the difference in logits assigned to the two names (*Mary* and *John*). We use a small dataset of 70 IOI examples created with Wang et al.'s [71] generator, as larger datasets did not provide significantly better results in our experiments and this size fit into GPU memory more easily.

---

[2]We exclude OLMo from our analysis due to missing checkpoints at the time of writing.

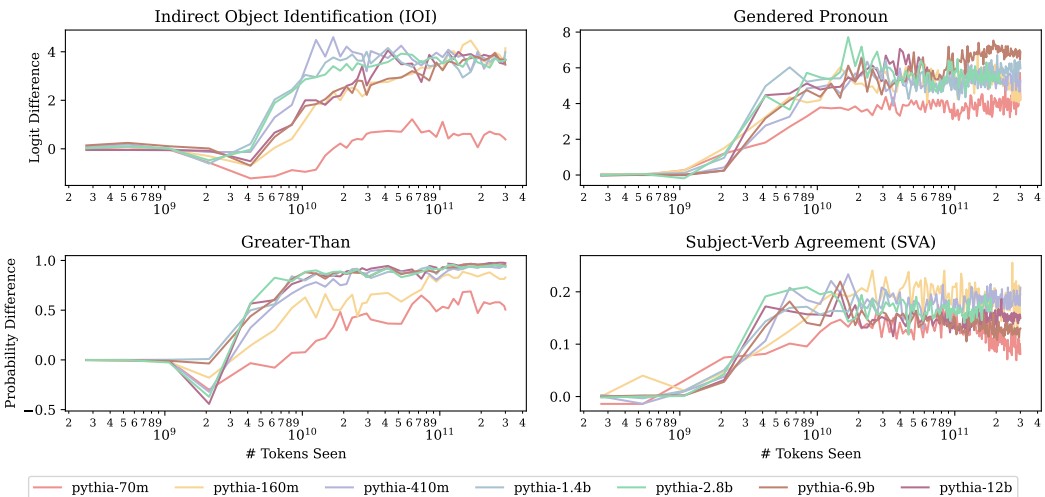

Figure 1: Task behavior across models and time (higher indicates a better match with expected behavior). Across tasks and scales, model abilities tend to develop at the same number of tokens.

**Gendered-Pronoun**    The Gendered-Pronoun task [69, 44, 15] measures the gender of the pronouns that models produce to refer to a previously mentioned entity. Prior work has shown "So Paul is such a good cook, isn't'"", models prefer the continuation "he" to "she"; we measure the degree to which this occurs via the difference in the pronouns' logits. In the corrupted case, we replace the "Jack" with "Mary"; we include opposite-bias examples as well. We craft 70 examples as in [44].

**Greater-Than**    The Greater-Than task [32] measures a model's ability to complete inputs such as $s =$"The war lasted from the year 1732 to the year 17" with a valid year (i.e. a year > 32). Task performance is measured via probability difference (prob diff); in this example, the prob diff is $\sum_{y=33}^{99} p(y|s) - \sum_{y=00}^{32} p(y|s)$. In corrupted inputs, the last two digits of the start year are replaced by "01", pushing the model to output early (invalid) years that decrease the prob diff. We create 200 Greater-Than examples with Hanna et al.'s [32] generator.

**Subject-Verb Agreement**    Subject-verb agreement (SVA), widely studied within the NLP interpretability literature [41, 52, 39], tasks models with predicting verb forms that match a sentence's subject. Given input such as "The keys on the cabinet", models must predict "are" over "is"; a corrupted input, "The key on the cabinet" pushes models toward the opposite response. We measure model performance using prob diff, taking the difference of probability assigned to verbs that agree with the subject, and those that do not. We use 200 synthetic SVA example sentences from [52].

Note that two tasks (IOI and Gendered-Pronoun) use logit difference as their metric, while the other two (Greater-Than and SVA) use probability difference. In general, we prefer to compute differences of logits rather than probabilities, as the former may be better at detecting important negative components when performing patching [75]. However, Greater-Than and SVA have multiple clean and corrupted answers, while logit difference, being unnormalized, is only compatible with one clean and corrupted answer. We thus compute a difference in probabilities instead, as in prior work.

## 3 Circuit Formation

### 3.1 Behavioral Evaluation

We begin our analysis of LLMs' task mechanisms over time by analyzing LLM behavior on these tasks; without understanding their task behaviors, we cannot understand their task mechanisms. We test these by running each model (Section 2.3) on each task (Section 2.4). Our results (Figure 1) display three trends across all tasks. First, all models but the weakest (Pythia-70m) tend to arrive at the same task performance at the end of training. This is consistent with our choice of tasks: they are simple, learnable even by small models, and scaling does not significantly improve performance.

Second, once models begin learning a task, their overall performance is generally non-decreasing, though there are minor fluctuations; Pythia-2.8b's logit difference on Gendered Pronouns dips slightly after it learns the task. In general, though, models tend not to undergo significant unlearning. The only marked downward trend (Pythia-70m at the end of SVA) comes from a weak model.

Finally, for each task we examined, we observed that there was a model size beyond which additional scale did not improve the rate of learning, and sometimes even decreased it; task acquisition appeared to approach an asymptote. We found this surprising due to the existence of findings showing the opposite trend for some tasks: [37, 57]. On some tasks (Gendered Pronouns and Greater-Than), all models above a certain size (70M parameters for Gendered Pronouns and 160M for Greater-Than) learn tasks at roughly the same rate. On IOI, models from 410M to 2.8B parameters learn the task the fastest, but larger models (6.9B and 12B) have learning curves more like Pythia-160m. We obtain similar results on more difficult tasks like SciQ [73]; for these results, see Appendix E.

What drives this last trend, limiting how fast even large models learn tasks? To understand this, we must delve into the internal model components that support these behaviors and trends.

## 3.2 Component Emergence

Prior work [54, 14, 63] has shown how a model's ability to perform a specific task can hinge on the development of certain components, i.e. the emergence of attention heads or MLPs with specific, task-beneficial behaviors. Prior work has also thoroughly characterized the components underlying model abilities two of our tasks, IOI and Greater-Than, at the end of training. We thus ask: is it the development of these components that causes the task learning trends we saw before? We focus on four main components, all of which are attention heads,[3] which we briefly describe here:

**Induction Heads** [54] activate on sequences of the form `[A] [B] ... [A]`, attending to and upweighting `[B]`. This allow models to recreate patterns in their input, and supports IOI and Greater-Than.

**Successor Heads** [29] identify sequential values in the input (e.g. "11" or "Thursday") and upweight their successor (e.g. "12" or "Friday"); this supports Greater-Than behavior.

**Copy Suppression Heads** [45] attend to previous words in the input, lowering the output probability of repeated tokens that are highly predicted in the residual stream input to the head. In the original IOI circuit, copy suppression heads hurt performance, downweighting the correct name. In contrast, we find (Appendix D) that they contribute positively to the Pythia IOI circuit by downweighting the incorrect name; this is possible because both names are already highly predicted in the input to these heads, and they respond by downweighting the most repeated one.

**Name-Mover Heads** [71] perform the last step of the IOI task, by attending to and copying the correct name. Unlike the other heads described so far, this behavior is specific to IOI-type tasks; their behavior across the entire data distribution has not yet been characterized.

Because the importance of these components to IOI and Greater-Than has been established in other models, but not in necessarily in those of the Pythia suite, we must first confirm their importance in these models. To do so, we find circuits for each model at each checkpoint using EAP-IG, as described in Section 2.2; note that circuits found at early checkpoints, where task performance is poor, are generally not meaningful. We omit Pythia-6.9b and 12b from circuit finding for reasons of computational cost. We then verify via path patching [28] that these component types appear within Pythia models' tasks circuits; see Appendices A and B for details on our methods and findings.

For each component, prior work has developed a metric to determine whether a model's attention head is acting like that component type; see Appendix D for details on these. Using these metrics, we score each of the heads in our models' circuits at each checkpoint, evaluating the degree to which each in-circuit head exhibits the aforementioned component behaviors. We then sum the score for each component across all in-circuit heads at each checkpoint, measuring how strongly the component behavior exists overall in the circuit at that checkpoint. We then normalize this sum across checkpoints, to understand how this behavior develops over time.

Our results (Figure 2) indicate that many of the hypothesized responsible components do emerge the same time as model performance increases. Most models' induction heads emerge soon after

---

[3]Though past work on e.g. Greater-Than studied task-relevant MLPs and neurons, what generalizable behavior they have is poorly understood, unlike the attention heads we study.

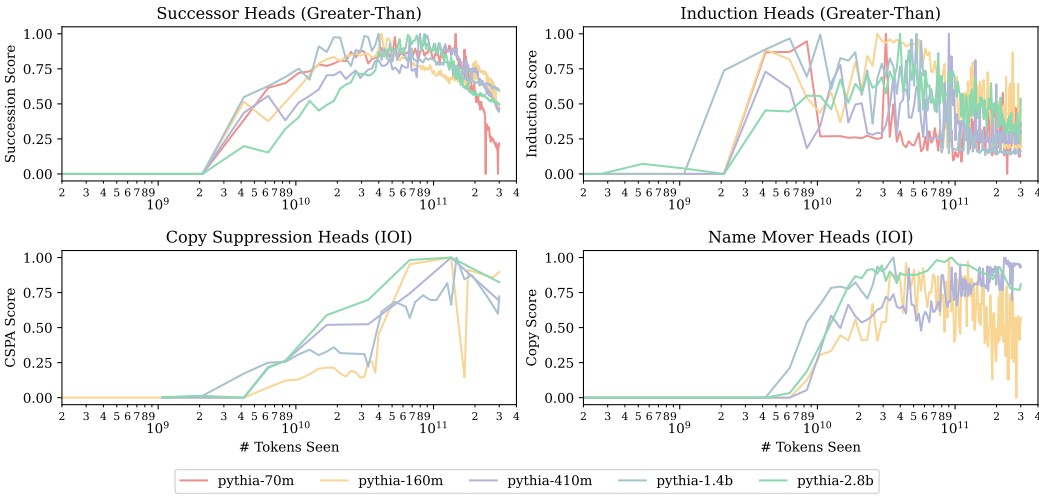

Figure 2: The development of components relevant to IOI and Greater-Than, across models and time. Each line indicates the degree to which attention heads in the circuit at each timestep exhibit the relevant component behavior. The timesteps at which component behavior emerges parallel those at which task performance emerges in Figure 1.

they have seen $2 \times 10^9$ tokens, replicating the findings in [54]; immediately after this, Greater-Than behavior emerges. The successor heads, also involved in Greater-Than, emerge in a similar timeframe. Curiously, the behavior of these heads eventually decreases, while task performance does not. We note that this could happen if the number of heads exhibiting successor behavior decreases, while the impact of said heads (on the model's logits) increases.

For IOI, the name-mover heads emerge at similar timesteps (2 - $8 \times 10^9$ tokens) across models, with a very high strength, during or just before IOI behavior appears. Copy suppression heads emerge at the same timescale, but at varying speeds, and with varying strengths. Given that these heads are the main contributors to model performance in each task's circuit, and they emerge as or just before models' task performance increases, we can be reasonably sure that they are responsible for the emergence of performance. The apparent ceiling on the speed at which these heads emerge, even for larger models, may be responsible for the same ceiling on how fast model performance emerges, underlain by these heads.

Throughout this section's analyses, we note an unusual trend: though model performance (Figure 1) does not decrease over time, the functional behavior of certain attention heads does. In the following section, we explore how this occurs.

## 4 Algorithmic Stability and Generalizability in Post-Formation Circuits

We demonstrated in Section 3 that across a variety of tasks, models with differing sizes learn to perform the given task after the same amount of training; this appears to happen because each task relies on a set of components which develop after a similar count of training tokens across models. However, in Figure 2, we observed that attention heads that performed a given functional behavior at a given point in training may later perform that behavior more weakly or not at all. This raises questions: when the heads being used to solve a task change, does the algorithm implemented by the model change too? And how do these algorithms generalize across model scale?

### 4.1 Model Behavior and Circuit Components Post-Formation

To understand how model components change over time, we analyze components and their algorithmic roles across training. We now zoom in on the components in one model, Pythia-160m, and study them over the course of training; where we earlier plotted only the top component (e.g. the top successor head), of each model, we now plot the top 5 of Pythia-160m's heads that exhibit a given functional behavior (or fewer, if fewer than 5 exist). By evaluating components and algorithms over

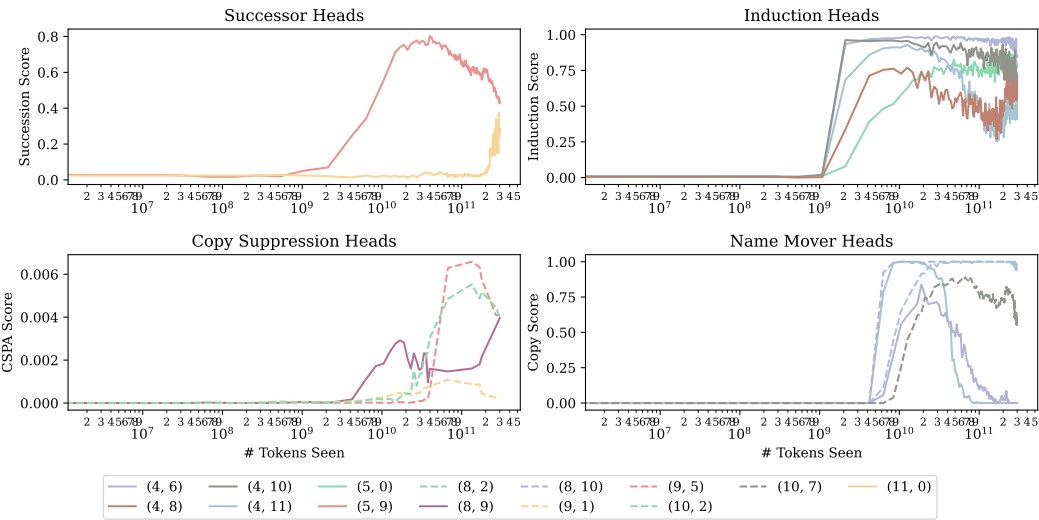

Figure 3: The development over time of components relevant to IOI and Greater-Than in Pythia-160m. Each line indicates the degree to which an attention head, denoted as (layer, head), exhibits a given function; higher values imply stronger functional behavior. Heads often lose their current function; as this occurs, other heads take their place.

Pythia-160m's 300B token training span, we extend beyond previous work, which studies models trained on relatively few ($\leq$ 50M) tokens [14, 63]; in such work, components and task behaviors appear constant after component formation.

By contrast, our results (Figure 3) show that over the longer training period of Pythia models, the identity of components in each circuit is not constant. For example, the name-mover head (4,6) suddenly stops exhibiting this behavior at $3 \times 10^{10}$ tokens, having acquired it after $4 \times 10^9$ tokens. Similarly, Pythia-160m's main successor head (5,9) loses its successor behavior towards the end of training; however, (11,0) exhibits more successor behavior at precisely that time. Such balancing may lead to the model's task performance remaining stable, as we observed in the prior section (Figure 1). It seems plausible that self-repair [46, 58] contributes to this behavioral stability, but we leave the question of the exact "load-balancing" mechanism to future work. Nevertheless, models can clearly compensate for losses of and changes in individual circuit components.

## 4.2   Circuit Algorithm Stability Over Training

This instability of functional components raises an important question—when attention heads begin or cease to participate in a circuit, does the underlying algorithm change? To answer this, we examined the IOI circuit, as it is the most thoroughly characterized [71] circuit algorithm of our set of tasks. Our investigation follows a three-stage approach: first, we analyzed the IOI circuit at the end of training, reverse-engineering its algorithm; next, we developed a set of metrics to quantify whether the model was still performing that algorithm; finally, we applied these metrics across checkpoints, to determine if the algorithm was stable over training.

At each checkpoint, we identified relevant heads through two criteria:

1. The head's effect on model performance when ablated via path patching
2. The head's score on component-specific functional tests (e.g., copy suppression score for name-mover heads)

Heads were included in our analysis only if they both: scored above a 10% threshold on their respective functional tests, and showed a negative effect on the logit difference when ablated. For S2-inhibition heads, we additionally verified that ablating positional information while preserving token information reduced both logit difference and name-mover head attention to the indirect object while increasing attention to the subject.

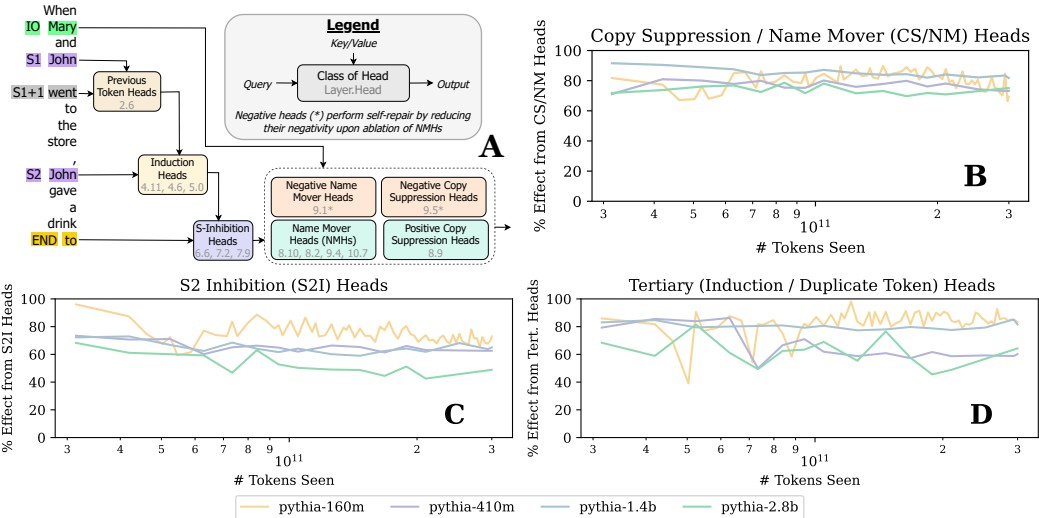

Figure 4: **A**: Pythia-160m's IOI circuit at the end of training (300B tokens). The remaining plots show the percent of model IOI performance that is explained by the Copy Suppression and Name-Mover Heads (**B**), the S-Inhibition Heads' edges to those heads (**C**), and the Induction / Duplicate Token Heads' connections to the S-Inhibition heads (**D**); higher percentages indicate that the corresponding edge is indeed important. Each of plots **B**-**D** verifies the importance of an edge from diagram **A**.

The first stage of our analysis is to analyze the IOI circuit at the end of training. Here, we present only the results of our analysis, but see Appendix B for details of this process, which follows the original analysis [71]. Figure 4**A** shows the circuit that results from our analysis; it involves three logical "steps," each of which involves a different set of attention head types. Working backwards from the logit predictions, the direct contributors towards the logit difference are name-mover heads and copy suppression heads. The former attend to the indirect object in the prompt and copy it to the last position; the latter attend to and downweight tokens that appear earlier in the input. In the next step, the name-mover heads (but not the copy-suppression heads) use on token and positional information output by the S-inhibition heads to attend to the correct token. Finally, S-inhibition heads rely on information from induction heads and duplicate-token heads.

Next, we quantify the extent to which the circuit depends on each of these three steps, via path patching [28]. When using path patching, we intervene on a node H (e.g., S-inhibition heads), route the change through an intermediate node or set of nodes R that depends on H (e.g., name-mover heads), and finally measure how the change in R affects the model output. For each step, our metric is the effect (on logit difference) of ablating the targeted components (through any intermediate nodes) divided by the effect of ablating all components that could affect R. For example, when measuring S-inhibition heads' importance to name-mover heads, the denominator is the ablation effect of all heads upstream of the name-mover heads. Higher ratios (0-100%) indicate greater relative importance of the targeted components. More details on path patching can be found in Appendix B, while details on the algorithm experiment itself can be found in Appendix A.

Finally, we compute each of these metrics for each model from 160M to 2.8B parameters in size.[4] We run them on each checkpoint post-circuit emergence (that is, when all component types appear in the circuit); for Pythia-160m, we test every checkpoint, and for the larger models we space out checkpoints to save compute, using approximately 1/3rd of the available checkpoints). We find (Figure 4**B**-**D**) that the behavior measured by these metrics is stable once the initial circuit has formed. Notably, in no model or metric are there dramatic shifts in algorithm corresponding to functional component shifts within the circuit. Moreover, all scores are relatively high, generally above 50%; the core solvers of the algorithm, copy suppression and name-mover heads, have scores above 70%. This suggests that analyses of circuits in fully pre-trained models may generalize well to other model states, rather than being contingent on the particular checkpoint selected.

---

[4]We omit Pythia-70m, as it does not learn the task; due to computational constraints, we omit Pythia-6.9b/12b.

Generalization across model scales also seems promising, as IOI circuit metrics from Pythia-160m are also high in larger Pythia variants. However, there is variation: while the name-mover, copy-suppression, and S-inhibition heads are at work in all models' circuits, the Pythia-160m circuit does not involve duplicate-token heads, while others do. So small differences exist amid big-picture similarity. Moreover, we stress that these algorithmic similarities might not hold for more complex tasks, for which a greater variety of algorithms could exist.

# 5 Graph-Level Circuit Analysis

We have now examined circuits over the training process from component and algorithmic perspectives. But how do the circuit subgraphs themselves change over time and scale? In Section 2.2, we explain how these subgraphs are collected; we applied this method to Pythia-70m through Pythia-2.8b for the tasks listed in Section 2.4. The result is a set of nodes and edges for each model, checkpoint, and task. Here, we briefly examine some trends we identified in the analysis of this data.

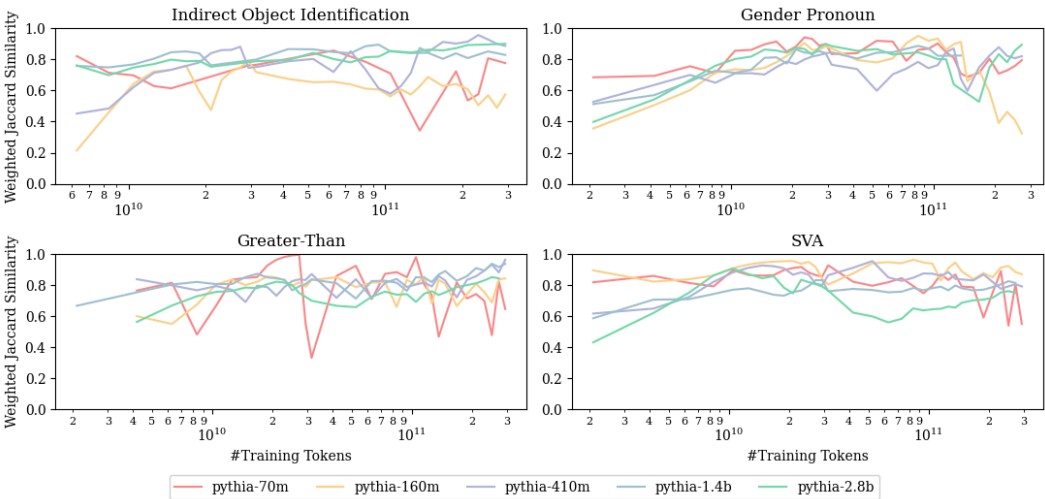

Figure 5: Exponentially-weighted moving average Jaccard similarity for circuit node sets over training token count. In general, larger models tend to have both higher average EWMA-JS and fewer abrupt fluctuations, indicating higher stability in the circuit constituents.

We first examine the consistency of the nodes in circuits over training. To measure this, we compute the Jaccard similarity (intersection over union) $x_t$ between the circuits at each given checkpoint and those at all previous checkpoints. In order to smooth out local fluctuations and observe longer-term trends, we apply an exponential weighting with a decay factor $\alpha = 0.5$, such that the value at a given checkpoint is the exponentially-weighted Jaccard similarity with the complete set of previous checkpoints. We calculate the weighted Jaccard similarity $\hat{x}_t$ at checkpoint $t$: $\hat{x}_t = 0.5\hat{x}_{t-1} + 0.5x_t$, Our results (Figure 5) suggest that larger models tend to form more stable circuits (with both higher average values and fewer sharp fluctuations); EWMA-Jaccard is more volatile for Pythia-70m/160m. In the Gendered-Pronoun circuit, we observe that significant changes can occur even late in training. We also computed how similar the circuit is during training with respect to the final circuit in Figure 7. we see more fluctuations, indicating that components swap during training at every checkpoint. Meanwhile, the upward trend indicates that circuits grow more and more similar to the final circuit during training.

We also compare the sizes (node counts) of the circuits over training. Across all four of our tasks, we find that the circuit size is positively correlated with the size of the models. We averaged node count across all the checkpoints for all the models on four tasks and calculated the pairwise correlation between the sizes of the models and the average node sizes. The Pearson correlation is $r = 0.72$ for IOI and SVA, 0.9 for Greater-Than, 0.6 for Gender Pronoun. We also find a high degree of variability—circuit sizes can remain stable or fluctuate significantly, with no clear pattern based on the model or the task. We leave further exploration of why this is the case to future work, but present our size metrics in Appendix C, Figure 6.

# 6   Discussion

**Implications for Interpretability Research**   While our findings are based on a limited set of circuits, they hold significant implications for mechanistic interpretability research. Our study was motivated by the fact that most such research does not study models that vary over time, like currently deployed models. However, the stability of circuit algorithms over the course of training suggests that analyses performed on models at a given point during training may provide valuable insights into earlier and later phases of training as well. Moreover, the consistency in the emergence of critical components and the algorithmic structure of these circuits across different model scales suggests that studying smaller models can sometimes provide insights applicable to larger models. This dual stability across training and scale could reduce the computational burden of interpretability research and allow for more efficient study of model mechanisms. However, further research is needed to confirm these trends across a broader range of tasks and model architectures.

**Quantization Model of Neural Scaling**   Our findings suggest evidence for the quantization model of neural scaling [49], which suggests that models learn tasks in order of decreasing use frequency. This model may explain why we observed models learning tasks (and sub-tasks, represented by circuit functional components) at similar times despite model size. As all Pythia models are trained on the same dataset, the same distribution of data (and thus task frequency) would be presented to each model, leading to acquisition of those common tasks.

**Limitations and Future Work**   Our analysis was limited to a narrow range of tasks feasible for small models. This limits in turn the scope of the claims that we can make. We believe it to be very possible that more complex tasks, not solvable by small models, which permit a larger range of algorithmic solutions, may show different trends from those that we discuss here. Such work would be valuable, though computationally expensive due to the model sizes required. Our analysis also studied models only from one model family, Pythia. It is thus not possible to tell if our results are limited to the specific model family we have chosen, which shares both architecture and training setup across model scale. Such work is in part hampered by the lack of large-scale model suites such as Pythia; future work could provide these suites to enable this sort of analysis. Finally, future work would do well to explore more complex phenomena, such as the self-repair and load-balancing mechanisms of LLMs, which ensure consistent task performance despite component fluctuations.

# 7   Related Work

**Interpretability Over Time**   LLMs' development over the course of pre-training has been studied with various non-mechanistic interpretability techniques, particularly behavioral interpretability, which characterizes model behavior without making claims about its implementation. Such longitudinal analyses have studied LLM learning curves and shown that models of different sizes acquire capabilities in the same sequence [74, 13], examined how LLMs learn linguistic information [72, 16, 12] and even predicted LLM behavior later in training [35, 4]. Nevertheless, behavioral studies alone cannot inform us about model internals. Prior work has studied the development of mechanisms in smaller models [51, 54], and suggests that model mechanisms can change abruptly, even as models' outward behavior stays the same. Other previous studies have examined the pre-training window where acquisition of extrinsic grammatical capabilities occurs [14].

**Mechanistic Interpretability**   We build on previous work in mechanistic interpretability, which aims to reverse engineer neural networks. *Circuits* are a significant paradigm of model analysis that has emerged from this field, originating with vision models [53] and continuing to transformer LMs [47, 71, 32, 68, 48, 40, 67]. Increasingly, research has tried to characterize the individual components at work within circuits, not only at the level of attention heads [54, 14, 63, 29, 45], but also neurons [69, 25, 59, 31, 70] and other sorts of features [9, 36, 43]. Recent work has also tried to accelerate mechanistic research via automated techniques [20, 6, 65, 33]. Though mechanistic interpretability is a diverse field, it is often tied together by a reliance on causal methods [69, 10, 26, 27, 47, 71, 11, 19], which provide more faithful mechanistic explanations.

## Acknowledgements

MH is supported in part by an OpenAI Superalignment Fellowship.

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

## A  Analysis of Task Circuits

### A.1  IOI Circuit & Algorithmic Criteria

To determine algorithmic consistency for the IOI circuit, we apply path patching as described in Appendix B in addition to using the component scores described in Appendix D. These are used to set thresholds for classifying attention heads. Though component score thresholds can be arbitrary, applying them consistently across all model checkpoints allows us to see the degree of similarity involved with model behavior.

Concretely, we use the following metrics and thresholds:

**Direct-effect heads** We initially perform path-patching on all model attention heads, measuring their impact on the logit different after the final layer of the model. We then classify attention heads as name-mover heads (NMHs), negative name-mover heads, and copy suppression heads (CSHs) based on copy score (for NMHs) or CPSA (for CSHs) of $> 10\%$, which yielded a small set of heads responsible for most of the direct effect. We measure the ratio of the absolute direct effect on logit difference for these heads vs. the total direct effect of all heads (including several unclassified heads) to obtain our first value.

Next, we conduct path-patching with NMHs as the receivers. This yields a set of heads that we then test for S2-inhibition (S2I) behavior, using Wang et al.'s [71] test for the effect of token signal vs. positional signal: does the ablation of these positional signal heads A). reduce the logit difference through the NMHs, B). reduce NMH attention (which determines what they copy) to the indirect object token, and C). increase attention to the subject tokens? If a head meets all of these conditions, we classify it as an S2I head, as it emits a signal used by the NMHs to decide what to copy. The total absolute effect of these heads on the NMHs is then divided by the total absolute effect of all heads on the NMHs, producing our second measurement.

Finally, we conduct path-patching with S2I heads as receivers. Here, we apply a simpler test since these heads can be quite diffuse throughout the model: Do the heads involved have above-average induction or duplicate-token scores? If so, we classify them as induction heads or duplicate token heads (confirming via manual examination of attention patterns and behavior), and divide the total absolute effect of these heads by the total absolute effect of all heads on the S2I heads, producing our third measurement.

These three metrics capture the extent to which known and classifiable model components contribute at each of the three primary levels of the IOI circuit. If the degree to which unknown or unclassified components contribute to any part of the circuit, we will see the corresponding score drop. As we see that in practice they tend to stay level, we conclude that there is a high degree of stability for this circuit.

## B  Other Circuit-Analysis Methods

Circuit analysis can be conducted via a number of different methods; the method used to find the original IOI circuit (and that we use to verify algorithmic consistency in this task) is Wang et al.'s [71] **path-patching**. Path patching is a specialized form of activation patching, used to isolate and analyze the influence of individual model components on a given task. Starting with two datasets (identical except for the key detail we want to base our circuit on, such as the correct and incorrect names in the IOI task), $x_{\text{orig}}$ and $x_{\text{altered}}$, where $x_{\text{altered}}$ is a counterfactual version of $x_{\text{orig}}$, the technique involves a sender attention head $h$ and a set of receiver nodes $R \subseteq M$ within the model's computational graph $M$. Initially, activations are recorded from both datasets. Subsequently, all attention heads except $h$ are locked to their activations from $x_{\text{orig}}$, while $h$ is updated with its activation from $x_{\text{altered}}$. This configuration allows for a forward pass on $x_{\text{orig}}$, capturing intermediate activations for nodes $r \in R$. A final forward pass on $x_{\text{orig}}$ then patches $R$ to these stored values, facilitating the assessment of $h$'s impact on the model's output.

Path patching aims to gauge the significance of the path $h \rightarrow r$ by comparing the model's logit differences across multiple pairs $(x_{\text{orig}}, x_{\text{new}})$. By averaging these differences over many pairs, the method effectively measures the impact of specific paths on model performance, providing insights into the contributions of individual components to the overall task. The process is iterative, such that

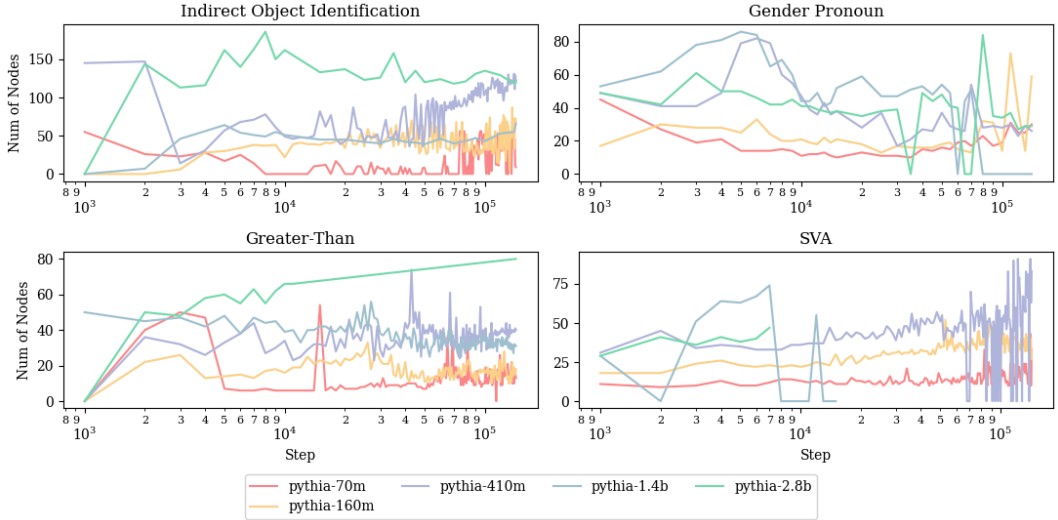

Figure 6: Number of Nodes in the circuits

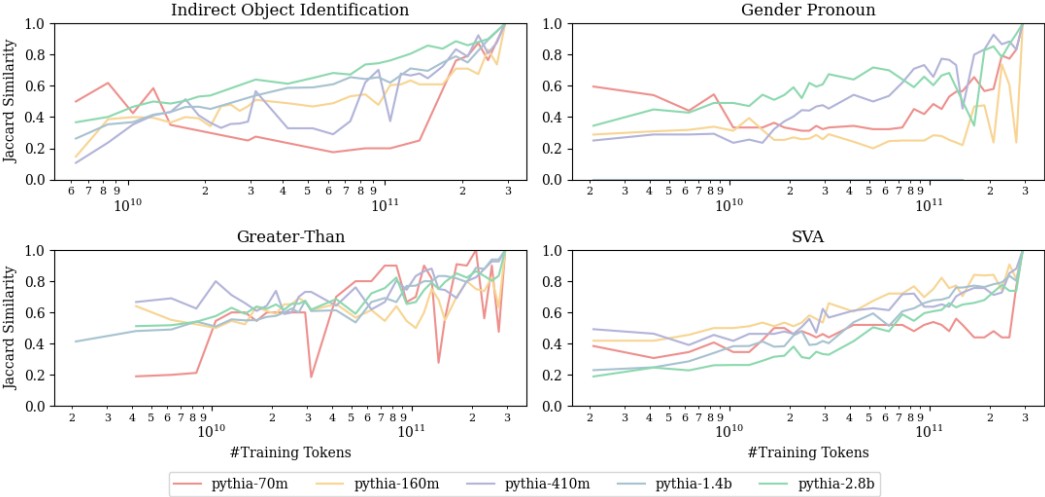

Figure 7: Node Jaccard Similarity of intermediate circuits with the circuit at the final checkpoint. Compared with the exponential weighted moving average (Figure 5, main text), we see more fluctuations, indicating that components swap during training at every checkpoint. Meanwhile, the upward trend indicates that circuits grow more and more similar to the final circuit during training.

a practitioner would start by observing which nodes impact the logits directly, and then proceeding backwards to see what nodes affect those first direct-effect nodes, and so on.

## C  Additional Size, Similarity & Change Rate Results

We graph out the number of nodes needed to generate faithful circuits across different checkpoints on all of the four tasks. Here we can observe that the number of nodes is positively correlated to the sizes of the models. When the model size increases the model needs more heads of the same kind to complete the same tasks. In the case of IOI, we can see the pink line for pythia-70m is at the bottom with the least number of nodes and the green line of 2.8b is at the top with the most number of nodes. This signifies a diffusion of the roles in attention heads. Increases in model size do not necessarily help heads become more specialized in their roles; rather, in these circuits more heads will take on the same roles.

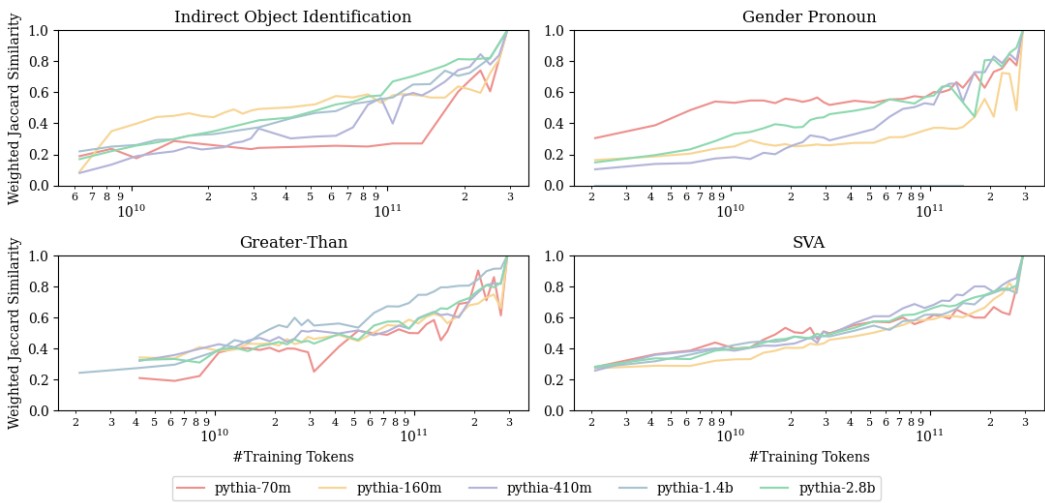

Figure 8: Edge Weighted Jaccard Similarity of the intermediate circuits with the circuit at the final checkpoint. The similarity slowly climbs and remains high at the end. Compared with the exact nodes, the weighted Jaccard Similarity of edges yields a smoother graph. It aligns with the conclusion that the model slowly drifts towards the final circuit along with components swapping during training

# D   Component Metrics

In this paper, we follow the metrics from previous literature in Wang et al. [71] for name-mover heads, McDougall et al. [45] for copy suppression heads, [54] for induction heads, and [29] for successor heads.

**Copy Score**   Following Wang et al. [71], we check if the Name Mover Heads copy over the names across training time by using the same metrics- **copy score**. To validate the Name Mover Heads, we studied what values are written via the head's OV matrix. We take the state of the residual stream after the first layer of MLP on the specific name tokens. Then we multiply it with the OV matrix of the given heads, multiplied with the unembedding matrix and also the final layer norm. This simulates what will happen if the head attended perfectly to that token. We define copy score as the proportion of samples that contain the input name token in the top 5 logits.

**CSPA Score**   McDougall et al. [45] introduced a novel approach named copy suppression-preserving ablation (CSPA), designed to ablate all behaviors of a specified attention head except for those related to copy suppression. This method involves two distinct types of ablation: OV ablation and QK ablation. In the OV ablation process, the output of an attention head at a destination token $D$ is represented as a weighted sum of result vectors from source tokens $S$, with the weights corresponding to the attention probabilities from $D$ to $S$ [22]. These vectors are then projected onto the unembedding vectors of their respective source tokens $S$, retaining only their negative components. Meanwhile, QK ablation involves mean-ablating the result vectors from each source token $S$, except for the top 5% of source tokens that are most likely to be predicted at the destination token $D$ based on the logit lens. For instance, in the phrase "All's fair in love and war," if the destination token $D$ is "and" and the token "love" is a highly predicted follower of $D$ and appears as a source token $S$, the result vector from $S$ is projected onto the unembedding vector for "love," and everything else is mean-ablated. This demonstrates how the attention head in question suppresses the prediction of "love." To evaluate the impact of the ablation, the token distribution output by the model for a given prompt ($\pi$) is compared with the distribution following an ablation ($\pi_{Abl}$) using KL divergence $D_{KL}(\pi||\pi_{Abl})$. By averaging these values over the OpenWebText dataset, $D_{CSPA}$ for CSPA and $D_{MA}$ for a mean ablation baseline are obtained. The proportion of the effect explained is then calculated as $1 - \frac{D_{CSPA}}{D_{MA}}$, with KL divergence chosen because a value of 0 indicates that the ablated and clean distributions are identical, implying that 100% of the head's effect is explained by the preserved components.

**Previous Token Score**   The Previous Token Score measures how effectively each attention head attends to the immediately preceding token. To compute this, we use a diagonal extraction on the attention pattern matrices, offset by one position. This captures the attention weights directed to the token that precedes each token in the sequence. The scores are averaged over all batches and tokens, providing a mean score for each attention head across all layers.

**Duplicate Token Score**   The Duplicate Token Score evaluates the propensity of each attention head to focus on duplicate tokens within a sequence. We achieve this by creating input sequences where the original tokens are repeated consecutively. The attention pattern matrices are then examined for their focus on tokens that are exactly a sequence length apart, indicating duplicate attention. The scores are calculated by averaging the attention weights along the specified diagonal, representing the attention paid to duplicate tokens.

**Induction Head Score**   Based on the prefix matching score described by Olsson et al. [54], the Induction Head Score is designed to assess the ability of attention heads to engage in induction, where they predict the next token in a repeated sequence based on previously encountered patterns. To measure this, we generate sequences where a segment is repeated and compute the attention pattern matrices. We extract the diagonals offset by one less than the sequence length, capturing the attention from the end of the first segment to the start of the repeated segment. The mean attention scores along this diagonal provide the Induction Head Scores, averaged over all batches and tokens.

**Succession Score**   The succession score [29] measures the degree to which an attention head performs succession, upweighting "2" in response to "1", or "May" given the input "April". As Gould et al.'s [29] code is not publicly available, we re-implement their successor score as follows. We create a dataset of successor, consisting of numbers (in digit and written form), days of the week, and months. Then, we perform the following procedure from [29]. Letting $W_E$ and $W_U$ denote the embedding and unembedding matrices of the model under study, $MLP_0$ denote the first (zero-indexed) MLP layer, and $W_{OV}$ be the $OV$ matrix of the head under study. Then $M = W_U W_{OV} MLP_0(W_E)$ is a square matrix whose size is that of the model vocabulary; each row thereof indicates, for the corresponding word $x$ in the vocabulary, the degree to which an output word $y$ is upweighted by the head under study, when $x$ is in the input. For each $(x, y)$ pair in our dataset (e.g. (3,4) or (Tuesday, Wednesday)) we verify that $M[x][y] > M[x][y']$ for all $y' \neq y$ in our dataset; that is, we ensure that the correct answer is more highly upweighted than any of the other possible answers in our dataset. The succession score is the proportion of examples in which that is the case.

## E   Additional Evidence for Task-Dependent Learning Ceilings

In addition to evaluations we performed ourselves, we also re-examined data collected during the Pythia training runs [5] on the SciQ [73], PIQA [8], WinoGrande [60], and ARC Easy [18] datasets. Each of these consist of a wide range of questions with multiple-choice answers, and accuracy was evaluated on the basis of the top choice logit produced by the model. We find that performance acquisition rates on these tasks followed the same pattern we detected with our simpler task datasets– that is, task learning rate seemed to approach an asymptote as the models increased in size. We describe the datasets below and present the results in Figure 9.

**SciQ**   The Science Questions (SciQ) dataset [73] consists of 13,679 crowdsourced multiple choice science exam questions ranging across physics, chemistry, biology, earth science, astronomy, and computer science. The questions cover a variety of complex reasoning skills such as causal reasoning, multi-hop inference, and understanding paragraph descriptions.

**PIQA**   The Physical Interaction Question Answering (PIQA) dataset [8] contains a total of 21k (across different subsets) multiple choice questions probing reasoning about basic physical commonsense knowledge. The questions test intuitive understanding of concepts like mass, volume, rigid objects, containment, stability, orientation, and more through grounded scenarios. Answering correctly requires applying physical reasoning.

**ARC Easy**   The AI2 Reasoning Challenge (ARC) dataset [18] is a collection of 7,787 multiple choice science exam questions compiled from various grade-level sources, including a research

partner of AI2. The questions cover diverse science topics and are structured as text-only prompts with 4 answer options. The ARC Easy subset consists of 5,197 of the relatively easier reasoning questions.

**Winogrande**    The WinoGrande dataset [60] was inspired by the original Winograd Schema Challenge (WSC) and consists of 44k problems generated through crowdsourcing and systematic bias reduction algorithms. Most of these are relatively easy for humans, but often difficult for LLMs.

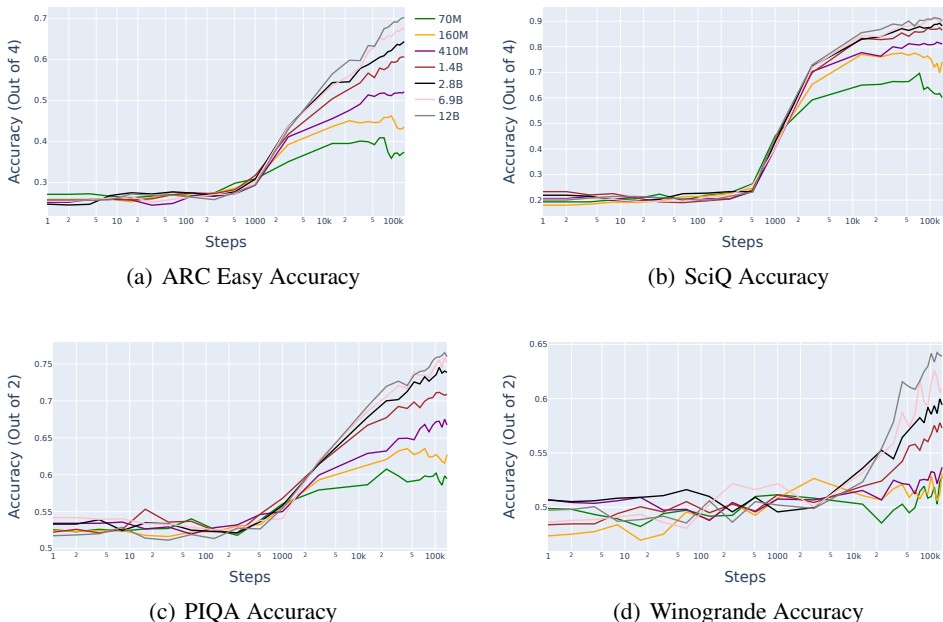

(a) ARC Easy Accuracy

(b) SciQ Accuracy

(c) PIQA Accuracy

(d) Winogrande Accuracy

Figure 9: Accuracy over training for four different datasets. Step numbers each represent approximately 2M tokens, so Step 1000 would be 2B tokens. We see that the rate of capability acquisition tends to approach an asymptote as models become larger.

## F   Compute

Experiments were conducted over two months a pod of 8 A40 GPUs, each with 50 GB of GPU RAM. As an upper bound, our experiments would require all of these GPUs to operate for a month to run all of our experiments, but in practice we did not require all GPUs running simultaneously. We estimate that 0.25 utilization of this pod would be required in practice to run these experiments.

## G   Licenses of Artifacts Used

The Pythia model suite is made available with an Apache 2.0 license. Wang et al.'s [71] IOI dataset and Newman et al.'s [52] SVA dataset are released under an MIT license. The remaining datasets (Greater-Than and Gendered-Pronouns) are released without any license specified.

