# OpenReview forum: "LLM Circuit Analyses Are Consistent Across Training and Scale"
_NeurIPS.cc/2024/Conference — NeurIPS 2024 poster_

### Official Review · Reviewer_v6cM · 2024-06-12

**Soundness:** 4
**Presentation:** 3
**Contribution:** 3
**Rating:** 5
**Confidence:** 4

**Summary:**

This paper examines how a few common circuits (IOI, greater than, verb agreement, gendered pronoun prediction) develop at different model scales and timings in training. The main findings are that these circuits develop at the same time across different model scales, that once they develop they do not disappear, and that although individual components of the circuits can change once developing the overall circuit structure mostly remains the same once it develops.

**Strengths:**

- The "sharing" behavior, where one attention head stops being suited for a particular purpose and another one starts being used for that purpose during training, is extremely interesting and to my knowledge a novel finding.
- The authors are the first to my knowledge to study how the structures of circuits change over training.
- The authors develop novel metrics that allow them to test if a given model implements a certain type of circuit.
- The authors contribute to universality by showing that previously studied circuits occurs in Pythia models of many scales.

**Weaknesses:**

- Only a few simple, well known circuits are analyzed. Significantly, the methods introduced are not scalable to large datasets of auto-discovered circuits.
- The motivation of fine-tuning/continual training is not quite applicable, since the work in this paper studies only pretraining checkpoints (which is fundamentally different from something like RLHF or fine-tuning on a narrow distribution).
- As the authors acknowledge, some results are not novel and merely reproduce earlier work (e.g. when induction heads arise).

**Questions:**

- Maybe I missed it, but what components are the circuits over? Are they MLPs + attention heads? Stating this very clearly could help, as well as pictures of the circuits in the appendix.
- It would be good to also have a graph showing Jaccard similarity of the current circuit vs. the final circuit, as it is hard to tell if the circuit is slowly drifting across nodes or if it stays mostly the same with a few heads switching back and forth.
- I'm confused about section 4.2. Are you rerunning the circuit discover algorithm at each checkpoint to get candidate heads? Or are you just running on all possible heads? How do you know the circuit uses the head if it’s the latter? I would also like to see Figure 4 at all tokens, including before the heads develop; since the lines are just horizontal it’s not clear to me that such heads don’t always exist, it'd be great to see a steep increase when the circuit develops.
- Line 162 says we're going to learn why bigger models do worse (an interesting finding in and of itself!) but I didn't see where this was touched on.
- One large claim (line 194) is that good performance on the circuits emerges at the same time as the heads emerge, but as these are on two different plots (figure 1 and figure 2) with different axes, this is hard to tell. Perhaps they could both be plotted on the same axes in the appendix with different line patterns to tell the different colors apart?
- Nit: in Figure 1, is there a reason some of it is logit diff and some of it is prob diff?
- The authors should ideally cite The Quantization Model of Neural Scaling (the hypothesis that language models learn skills ("quanta") in order of importance), since their work supports the Quanta hypothesis in a very interesting way (the circuits arise at the same time across models).
- For weighted jaccard similarity, it would be good to “see” attention heads switching. Maybe you can use the weight of edges in the DAG to assign nodes importance and do some sort of importance-weighted jaccard? As is it is hard to tell what’s changing in the circuits, much of the variance in the graph could just be from the size of the circuit changing as opposed to the more interesting "sharing".

I would be happy to raise my score if some of these questions/concerns were addressed.

**Limitations:**

Yes.

---

> ### Author Rebuttal · Authors · 2024-08-07
>
> Thanks for your thorough review! We respond below, abbreviating your comments for space reasons.
> ## Weaknesses
>
> > the methods introduced are not scalable to large datasets of auto-discovered circuits
>
> We disagree in part about the scalability of our methods. EAP-IG is efficient, and can now be used with models around 7B parameters in size (unfortunately, this was not the case when writing the paper). The fact that small models are increasingly capable means that studying e.g. OLMo-1B or 7B could show us how models learn even rather complex tasks. However, we do agree that studying 156 checkpoints for each model was compute-intensive, and some parts (copy suppression evaluation) scale poorly.
>
> > the motivation of fine-tuning/continual training is not quite applicable
>
> We agree that narrow fine-tuning doesn’t not quite fit as a motivation, and will revise this. However, continuous training is often done on a wider distribution of text (e.g., a large variety of user queries) and there may also be cases where researchers would want to look at circuits at intermediate checkpoints of a model before it has finished training.
>
> > some results are not novel and merely reproduce earlier work
>
> Although our induction head findings do reproduce findings in Olsson et al., we primarily mention them in passing; the bulk of our paper consists of novel findings: other component types, performance emergence, algorithmic stability. We will try to focus on the more novel parts of our work.
>
> ## Questions
> > what components are the circuits over?
>
> The circuits are composed of attention heads and MLPs (plus the inputs and logits). We will clarify this!
>
> > It would be good to also have a graph showing Jaccard similarity of the current circuit vs. the final circuit
>
> Thanks for the suggestion, implemented in Figures 2/3 of the response PDF. We see that in Figures 2/3 there are still many fluctuations compared with the final circuits which indicates that component swapping occurs during training. We can also observe a rising trend, indicating that the circuit grows increasingly similar to the final circuit during training. Therefore, the model both slowly drifts towards the final circuit, even as components swap during training.
>
> > I'm confused about section 4.2. Are you rerunning the circuit discover algorithm at each checkpoint to get candidate heads? Or are you just running on all possible heads? How do you know the circuit uses the head if it’s the latter? I would also like to see Figure 4 at all tokens, including before the heads develop.
>
> Apologies for the confusion in Section 4.2! To clarify, there are two criteria for including a head in a group: whether it meets a component score threshold, and whether it is important to model performance as measured by the path patching causal intervention. The latter intervention is roughly what EAP-IG approximates, so it’s actually a stronger test than checking circuit membership; see our response to dE5s for more details. Given this procedure, we start Figure 4 at the 30B token mark (10% of training) because this is where name-movers (NMHs) first appear as part of the IOI circuits. S2-inhibition and other heads are found via their impact on NMHs, so without NMHs, there’s nothing to measure / plot. Before the 30B token mark, most IOI circuits in these models consist simply of a single copy suppression head. Gradual emergence of relevant component types is better seen in Figures 2 and 3 of the main text.
>
> > Line 162 says we're going to learn why bigger models do worse
>
> Our apologies for the confusion! Large models do not learn any faster than smaller models do (above a certain threshold), because such models do not develop the responsible heads any faster than smaller models do. We don’t say that explicitly right now, but we will add that to the paper.
>
> > Perhaps they could both be plotted on the same axes in the appendix with different line patterns to tell the different colors apart?
>
> We agree that plotting things in one plot, with the same axis would help, but we didn’t have the space in our response PDF. However, we plotted the sum of component effects for in-circuit heads using the same x-axis as the behavioral plot. In this new plot, you can see clearly how greater-than components arise at or just before task behavior emerges, at 2x$10^9$ tokens. For IOI, copy suppression, the initial part of the circuit, arises at 4x$10^9$ tokens, while the name movers form a bit later, at 7 to 8x$10^9$ tokens. We will include the plot that you suggest in the appendix.
>
> > in Figure 1, is there a reason some of it is logit diff and some of it is prob diff?
>
> For tasks with one in/correct answer, we use logit diff; for tasks with multiple, we use prob diff, aligning with the original metrics for these tasks. Logit diff doesn’t quite work for multi-answer tasks; we can’t sum over the logits of all in/correct answers as with prob diff. We could use prob diff for all tasks, but some argue against this (Heimersheim and Nanda, 2024).
>
> > The authors should ideally cite The Quantization Model of Neural Scaling
>
> Thank you for pointing out that connection! We’ll add that citation.
>
> > Maybe you can use the weight of edges in the DAG to assign nodes importance and do some sort of importance-weighted jaccard?
>
> We have included related plots in the response PDF (Figures 2 and 3). For now, instead of plotting importance-weighted Jaccard similarity (JS) for node sets, we do the same for edge sets, as EAP-IG gives us only edge importance; we think computing node importance from edge importance would give similar results. We compute the Jaccard similarity between intermediate checkpoints and final circuit. We can see that the JS slowly climbs and remains high at the end. It aligns with the conclusion that the model both slowly drifts towards the final circuit along with components swapping during training. To see specific attention head switching, we refer to Figure 3 (main text).

---

> > ### Comment · Reviewer_v6cM · 2024-08-10
> >
> > Re scalability, I meant that to run the same analysis on a new circuit, you need to figure out a custom test that will identify each head of that circuit, which is very non trivial (correct me if I'm wrong). This to me feels like a large weakness in being able to scale up this analysis to a large database of circuits and reproduce the results there.
> >
> > Thank you for plotting the additional Jaccard similarity graphs! I think the results of this experiment were quite nice. I am curious how you think the new graphs affect the discussion of load balancing. It seems there are now fewer spikes; does this mean that the fluctuations might mostly be due to noise in the interventions in the less important edges?
> >
> > To make sure I understand 4.2, you're trying to find out if the circuit "works" in a different way in different points in time? I am a little uncomfortable still with this experiment, it seems a bit too much to me like you are assuming the conclusion (that the circuit works in a certain way) and then showing that when you assume that conclusion and ablate those edges, then model performance decreases. The mere fact that you can find the heads at all seems to me that you are assuming the circuit works in a certain way, and then the ablation doesn't seem necessary. But please correct me if I am wrong.
> >
> > For now, I will keep my score, thank you for responding to my comments!

---

> > > ### Author Response · Authors · 2024-08-12
> > >
> > > Thanks for your response; it’s much appreciated! As for the points you make:
> > > > Re scalability, I meant that to run the same analysis on a new circuit, you need to figure out a custom test that will identify each head of that circuit, which is very non trivial (correct me if I'm wrong). This to me feels like a large weakness in being able to scale up this analysis to a large database of circuits and reproduce the results there.
> > >
> > > While EAP-IG doesn’t require custom tests (as it is agnostic regarding the nature of the sub-functions in the circuit), we do agree that the fact that there exist so few well-characterized attention heads makes testing head identity difficult. However, this isn’t unique to our method: automatically assigning / verifying the semantics of model components is one of the big challenges of mechanistic interpretability. The difficulty of this challenge is why we don’t yet have a large database of well-understood circuits to scale to. However, we hope that as that nascent line of research progresses, more heads (and tests for them) will emerge! Finding circuit structure was also very slow just two years ago; ideally,the rate at which we understand circuit semantics will also accelerate. Once these tests emerge, it would be relatively straightforward to repeat our experiments with them.
> > > > Thank you for plotting the additional Jaccard similarity graphs! I think the results of this experiment were quite nice. I am curious how you think the new graphs affect the discussion of load balancing. It seems there are now fewer spikes; does this mean that the fluctuations might mostly be due to noise in the interventions in the less important edges?
> > >
> > > We’re glad our new graphs proved useful! In spot-checking the data points individually, we do indeed see what you suggested: that fluctuations occur more frequently in less-important edges. (Though, we do emphasize that even highly-important nodes also drop out of the circuit at times, as per Figure 2 in the main paper).
> > > > To make sure I understand 4.2, you're trying to find out if the circuit "works" in a different way in different points in time? I am a little uncomfortable still with this experiment, it seems a bit too much to me like you are assuming the conclusion (that the circuit works in a certain way) and then showing that when you assume that conclusion and ablate those edges, then model performance decreases. The mere fact that you can find the heads at all seems to me that you are assuming the circuit works in a certain way, and then the ablation doesn't seem necessary. But please correct me if I am wrong.
> > >
> > > We understand your concern about 4.2, but want to point out here that our approach is slightly different than you suggest. Essentially, we are trying to test whether our assumptions about the circuit are correct. For example, for Figure 4B, we first identify all CS/NM heads in the model that are contributing positively to performance (without assuming the overall algorithm), and our metric is the ratio between “performance reduction if *only* CS/NM heads are ablated” (which is our hypothesis) and “performance reduction if *all* heads are ablated,” finding that this ratio is quite high, confirming our hypothesis that they are important.
> > >
> > > Having found that NM heads exist at all checkpoints in these models, we then identify S2 inhibition heads through their effect on those NM heads, and our next metric in Figure 4C is the ratio between “performance reduction if only S2I heads are ablated, as intermediated through the NM heads” and “performance reduction if all heads upstream of NM heads are ablated, as intermediated through the NM heads” (and then so on for Figure 4D). There are things that these tests don’t cover, like the relative importance of CS vs. NM heads, or induction vs. duplicate token heads, or what the non-S2I heads do to affect the NM heads, but our tests do provide evidence that specific key algorithmic steps are occurring consistently in these models.

---

### Official Review · Reviewer_wt3i · 2024-07-03

**Soundness:** 3
**Presentation:** 4
**Contribution:** 2
**Rating:** 7
**Confidence:** 4

**Summary:**

This paper studies the stability of circuits in language models during pre-training and across different model sizes. Specifically, it examines the Pythia model suite and a selection of four simple tasks with known circuits: indirect object identification (IOI), subject-verb agreement, greater-than, and gendered pronouns. The analysis involves multiple steps, each repeated across different model sizes (70m - 2.8b) and pre-training checkpoints:

1. Evaluate the ability of the language model to solve each of these tasks.
2. Evaluate the emergence of specific attention heads, which previous work has established as important for each of these tasks.
3. Narrow down on the IOI task to evaluate whether the three core algorithmic steps, again known from previous work, are consistent across training and scale. To this end, the authors use path patching to ablate the connection between the components involved in each of these steps.
4. Evaluate the consistency of the circuits during training by looking at component and edge overlap. To identify these circuits, the authors use edge attribution patching.

Overall, the results reveal a significant level of consistency of circuits during training and across scale: important attention heads tend to start emerging after roughly the same number of tokens, although at different paces; the effect of important heads in the IOI circuit is somewhat consistent once these components emerge; and there is significant circuit component overlap across checkpoints.

**Strengths:**

- The paper investigates an interesting question: to what extent might existing circuit analysis results generalise across training and scale? Understanding the training dynamics of circuits is important for the field of (mechanistic) interpretability.
- The authors employ a variety of analysis techniques, including path patching and edge attribution patching, to systematically and causally test their hypotheses.
- Despite focusing on a narrow set of tasks, the authors identify potentially general motifs, such as “load balancing”. These insights open up avenues for future work to develop a more fundamental understanding of the stability of circuits across training and scale.

**Weaknesses:**

- While the results demonstrate a significant degree of consistency and stability of circuits across training, the focus on a small number of simple tasks provides limited insight into whether circuits for other tasks are consistent as well.
- The results reveal several training dynamics that are left unexplored. For example, load balancing or the observation that many attention heads emerge after roughly the same number could have been explored in more detail. I believe a detailed investigation of one of these phenomenons could have significantly improved the paper.

Minor issues:
- The title of the paper (“Stability and Generalizability of Language Model Mechanisms Across Training and Scale”) does not match the title in OpenReview (“LLM Circuit Analyses Are Consistent Across Training and Scale”).
- Typo in L67: “… we can verify that they are …”

**Questions:**

1. In Figure 5, you focused on the Jaccard similarity with *all* previous checkpoints. This makes it hard to evaluate whether circuits had phases of high consistency with only the previous checkpoint. Did you observe any periods of significant circuit stability during training? This would be interesting to see as the authors of latent state models of training dynamics [1] suggest phase transitions between different algorithmic solutions.

2. You mention that “Circuits in larger models require more components, with circuit sizes positively  correlating with model scale” in your contributions, but I don’t see this discussed in the following sections. Have you been able to study how these circuits differ across scale? Do we find duplications of the same components? How does this relate to the higher stability of circuits in larger models?

[1] M. Y. Hu, A. Chen, N. Saphra, and K. Cho, ‘Latent State Models of Training Dynamics’, arXiv [cs.LG]. 2024.

**Limitations:**

The selection of tasks limits the generalisation of their findings, as previous studies suggested that various behaviours of language models emerge at scale [2] or qualitatively change across scale [3]. Both of these suggest that circuits still fundamentally change for more complex tasks. However, I believe that this is mostly addressed in the limitations section of the paper.

[2] J. Wei et al., ‘Emergent Abilities of Large Language Models’, arXiv [cs.CL]. 2022.

[3] J. Wei et al., ‘Larger Language Models Do In-Context Learning Differently’, arXiv [cs.CL]. 2023.

---

> ### Author Rebuttal · Authors · 2024-08-07
>
> Thanks for  your helpful review! We agree that many of the issues you bring up are important, and have attempted to address them below.
>
> ## Weaknesses:
> > While the results demonstrate a significant degree of consistency and stability of circuits across training, the focus on a small number of simple tasks provides limited insight into whether circuits for other tasks are consistent as well.
>
> We agree that it remains to be seen whether more complex circuits, or a broader set of circuits in general show the same stability over training and scale. Currently, however, only a very small set of circuits have yet been identified, and finding new circuits was somewhat beyond the scope of our paper. We hope that our work will be useful to follow-on investigations of future circuits and their potential for consistency across these dimensions. We do not claim that all circuits will retain this consistency, and it remains to be seen what happens in much, much larger models; however, we do suggest that this provides some evidence that circuits found at specific points and model sizes can provide information that holds to at least some degree in across these dimensions.
>
> > The results reveal several training dynamics that are left unexplored. For example, load balancing or the observation that many attention heads emerge after roughly the same number could have been explored in more detail. I believe a detailed investigation of one of these phenomena could have significantly improved the paper.
>
> We also agree that investigating load balancing or token-dependent formation of components would have added to the paper; however, we felt such investigations likely would deserve their own dedicated projects (likely involving significant amounts of model training to get more granular checkpoints) and research output, and as such decided the scope of this paper would be best limited to an initial investigation of a wide set of phenomena. Those two topics are certainly worth investigation, however, and we hope follow-up work will address them!
>
> >The title of the paper (“Stability and Generalizability of Language Model Mechanisms Across Training and Scale”) does not match the title in OpenReview (“LLM Circuit Analyses Are Consistent Across Training and Scale”).
>
> > Typo in L67: “… we can verify that they are …”
>
> Thanks for catching these! We will fix them.
>
> ## Questions:
> > 1. In Figure 5, you focused on the Jaccard similarity with all previous checkpoints. This makes it hard to evaluate whether circuits had phases of high consistency with only the previous checkpoint. Did you observe any periods of significant circuit stability during training? This would be interesting to see as the authors of latent state models of training dynamics [1] suggest phase transitions between different algorithmic solutions.
>
> To provide more detail the question on Jaccard similarity and circuit stability, we have added a set of graphs (Figures 2 and 3) that show circuit Jaccard similarity at each checkpoint to the final circuit. We do observe periods of stability in some models and tasks, but in other cases we see gradual or spiky transitions towards the final circuit. We also note that we conducted EAP-IG on a number of seed variations of the Pythia models, finding a variety of patterns; in some models, the circuits were quite stable for long periods, while others showing oscillating sharp changes between checkpoints. Unfortunately, we didn’t have room to include these plots, but we could include them in the appendix.
>
> > 2. You mention that “Circuits in larger models require more components, with circuit sizes positively correlating with model scale” in your contributions, but I don’t see this discussed in the following sections. Have you been able to study how these circuits differ across scale? Do we find duplications of the same components? How does this relate to the higher stability of circuits in larger models?
>
> In Section 5, we compute the Pearson correlation between circuit sizes and model sizes. The Pearson correlation is r = 0.72 for IOI and SVA, 0.9 for Greater-Than, 0.6 for Gender Pronoun.
>
> We do find some differences in circuits across model scales. As we cover in Section 4, the algorithmic structure remains similar, but as shown in Section 5, the circuits are larger in larger models. In the case of the IOI task, some components demonstrate more replication than others; for example, copy suppression heads and S-inhibition heads remain few in number across circuits in different model scales, while induction heads and name-mover heads increase significantly. On balance, less-volatile head types seem to increase the most in number, which does indeed support your suggestion that this is related to circuit stability in larger models. Regardless, though, all identified components seem to be present in all models, and seemed to be similarly important to their circuits for IOI.

---

> > ### Comment · Reviewer_wt3i · 2024-08-11
> >
> > Thank you for the additional information regarding differences in circuits across model scales and the additional Figures 2 & 3. I appreciate the effort and believe that these results add significant nuance to your findings. Thus, I strongly suggest including them in the final version of the paper to enhance the clarity of your analysis. I think the other reviewers have brought up some valid concerns (e.g. generality of findings), but I believe the authors have done an adequate job at trying to address them. Overall, I believe this paper would be an interesting contribution to the conference and will increase my score to accept.

---

### Official Review · Reviewer_865w · 2024-07-05

**Soundness:** 3
**Presentation:** 3
**Contribution:** 4
**Rating:** 6
**Confidence:** 4

**Summary:**

This study explores the emergence and evolution of internal mechanisms in language models of varied sizes during the training process. Specifically, it examines simple tasks such as IOI, Greater-than, Gender Pronoun, and Subject-verb agreement using Pythia models. The findings indicate that models of different sizes tend to learn these tasks after a similar number of token counts during training. Moreover, while individual components of the models may change in functionality, the overall algorithm implemented by the models remains consistent throughout the training process. Lastly, the study identifies that once a circuit emerges, it generally remains stable thereafter.

**Strengths:**

1. This work is highly relevant for two key reasons:
   - Most mechanistic interpretability studies do not examine the internal mechanisms of models throughout the training and fine-tuning processes. As a result, they fail to offer a comprehensive understanding of how and when a model learns its mechanisms and how these mechanisms evolve after their creation.
   - Unlike many existing studies, this work analyzes multiple tasks across various models of different sizes.
2. The results from Section 4 indicate that, although some components of the model change their functionality, the overall algorithm used by the model to solve simple tasks, such as IOI, remains consistent. This finding aligns with previous results from [1], which state that even though fine-tuned models have larger circuits, their mechanisms remain consistent, even for more complex tasks like entity tracking.
3. The presentation is clear, but it does require the reader to have some background in mechanistic interpretability. Additionally, there are a few other pertinent works on mechanistically understanding the impact of fine-tuning that the authors should cite [2, 3].


[1] Prakash et al, "Fine-Tuning Enhances Existing Mechanisms: A Case Study on Entity Tracking", 2024.

[2] Jain et al, “Mechanistically analyzing the effects of fine-tuning on procedurally defined tasks”, 2023.

[3] Lee et al, “A Mechanistic Understanding of Alignment Algorithms: A Case Study on DPO and Toxicity”, 2024.

**Weaknesses:**

1. I’m unsure about the results of section 3.2 stating that emergence of circuit components that are involved in the internal computation of the tasks at the same time as the model behavior performance, suggests that the former is responsible for the emergence of latter, because of the following reasons:
   - All the model heads are evaluated, rather than only those involved in the circuit performing the task. It is possible that some heads exhibit certain behaviors without actually being part of the circuit. Thus, concluding that their occurrence is responsible for the model's performance could be misleading.
   - Furthermore, the analysis is primarily conducted for a single task, IOI, rather than all four tasks mentioned earlier. Although some of the heads studied are involved in the Greater-than tasks, MLP neurons, which have been shown to be part of the circuit, are not analyzed.
2. While it is interesting to note that individual circuit components emerge simultaneously with the model's behavior, it still does not explain why these components emerge after similar token counts in models of varied scales.
3. Section 4 investigates only the IOI task, which raises concerns regarding the generalizability of its results.

**Questions:**

1. The finding that models of varied sizes learn a task after a similar number of tokens is intriguing and somewhat counterintuitive. This makes me wonder if analyzing how gradient updates modify model weights and comparing these changes across models could provide a better understanding of how language models learn (potentially in future works).
2. Section 3.2 states that to validate the importance of four mentioned types of attention heads, a circuit is discovered for each model at each checkpoint. I’m unsure of what a circuit discovery algorithm will discover for early checkpoints where the model does not even have behavioral ability to perform the task. So, it’s surprising to me that authors were still able to identify circuits and functionality of their components which is consistent with existing discovered circuits in the literature. I would like to see evaluation results of these circuits.
3. Section 2.2 mentions that there is no definitive method for verifying the entirety of the identified circuit. This needs more explanation,  particularly regarding why metrics like completeness proposed in [4] are considered inadequate for this purpose.

[4] Wang et al, “Interpretability in the Wild: a Circuit for Indirect Object Identification in GPT-2 small”, 2022.

**Limitations:**

The primary limitation of this work is the use of simple tasks such as IOI, Greater-than, Subject-verb, and Gender Pronoun for analysis. Some of the results may not be applicable to more complex tasks. Additionally, relying solely on Pythia model suits trained using the same training data poses a risk to the generalizability of the findings. However, despite these limitations, the results are insightful and should be valuable for the mechanistic interpretability community and beyond.

---

> ### Author Rebuttal · Authors · 2024-08-07
>
> Thanks for your insightful review! These are good points, which we answer below.
>
> ## Weaknesses
> > 1.1 All the model heads are evaluated, rather than only those involved in the circuit performing the task. It is possible that some heads exhibit certain behaviors without actually being part of the circuit. Thus, concluding that their occurrence is responsible for the model's performance could be misleading.
>
> Indeed, this could be a concern. In our PDF response, we plotted the sum of all component effects from all heads in the circuit (Figure 1). We see that the emergence of these components comes right as (or just before) the emergence of task behavior as well, indicating that our previous results are not due to the inclusion of non-circuit heads.
>
> > 1.2 Furthermore, the analysis is primarily conducted for a single task, IOI, rather than all four tasks mentioned earlier. Although some of the heads studied are involved in the Greater-than tasks, MLP neurons, which have been shown to be part of the circuit, are not analyzed.
>
> It’s true that Hanna et al. (2023) study MLPs and their neurons. However, unlike the induction and successor heads on which we focus, these aren’t known to perform a generalizable function that is reused across tasks. We could track the MLPs that connect to the logits in the circuit, and which neurons therein matter most, but we wouldn’t gain any insights about how higher-level model abilities develop. For these reasons, we excluded them.
>
> > 2. While it is interesting to note that individual circuit components emerge simultaneously with the model's behavior, it still does not explain why these components emerge after similar token counts in models of varied scales.
>
> We agree that the phenomenon of components emerging at similar token counts across model scales deserves more exploration. A deeper look into this phenomenon seemed beyond the scope of our paper, but we hope it will be the subject of future work. We hypothesize that the formation of the other head types may occur in a phase change similar to the induction head phase change observed in Olsson et al. (2022), though model retraining for the purpose of obtaining more granular checkpoints would be needed to look in further detail at what is happening as these heads develop. This by itself would not explain why this happens, however, and we hope to see further investigations with additional approaches.
>
> > 3. Section 4 investigates only the IOI task, which raises concerns regarding the generalizability of its results.
>
> It is true that running a similar analysis on a broader set of circuits would strengthen our claim of algorithmic consistency. Unfortunately, few circuits with a clear and quantifiable algorithm beyond IOI have been identified to-date; even quantifying the relatively well-characterized Greater-Than is challenging. For this reason, we used IOI as the subject of our analysis. We do not claim that the property of generalizability over time and scale will apply to all circuits, especially more complex ones, but we hope this is investigated further.
>
> ## Questions
> >1. The finding that models of varied sizes learn a task after a similar number of tokens is intriguing and somewhat counterintuitive. This makes me wonder if analyzing how gradient updates modify model weights and comparing these changes across models could provide a better understanding of how language models learn (potentially in future works).
>
> We have also hypothesized that there may well be a connection between the number of gradient updates and the magnitude of changes to model weights required for models to learn a task. This isn’t quite in the scope of the current work, but we agree this is an interesting question.
>
> >2. Section 3.2 states that to validate the importance of four mentioned types of attention heads, a circuit is discovered for each model at each checkpoint. I’m unsure of what a circuit discovery algorithm will discover for early checkpoints where the model does not even have behavioral ability to perform the task. So, it’s surprising to me that authors were still able to identify circuits and functionality of their components which is consistent with existing discovered circuits in the literature. I would like to see evaluation results of these circuits.
>
> This is true—we run the circuit at all checkpoints, but at early checkpoints, there’s little model behavior to localize. The circuits discovered before model behavior emerges are not very meaningful; though they attain 80% of the model’s original performance, that performance is near 0. These pre-performance “circuits” consist of fairly stochastically-changing, hard-to-interpret sets of components that contribute only to the extremely small positive and negative variances in the task performance metrics. Due to this, we don’t base our conclusions on these early circuits; we will stress this in the text.
>
> > 3. Section 2.2 mentions that there is no definitive method for verifying the entirety of the identified circuit. This needs more explanation, particularly regarding why metrics like completeness proposed in [4] are considered inadequate for this purpose.
>
> Wang et al. (2023) indeed propose a completeness metric, which involves comparing model and circuit performance while ablating a set of components from both. However, this has flaws. For one, it’s challenging to choose components; Wang et al. propose three methods for doing this, which yield different results, and are not all compatible with our needs. More importantly, these ablations are computationally expensive, and not feasible to perform over many checkpoints. As a result, this metric has not been widely adopted, and we find most of the proposed alternatives (e.g. faithfulness of the circuit's complement) inadequate. We can add some discussion of this issue.

---

> > ### Comment · Reviewer_865w · 2024-08-09
> >
> > Thank you for the response. I believe including some of these points in the manuscript will improve its clarity. I will be keeping my original score.

---

> > > ### Author Response · Authors · 2024-08-13
> > >
> > > Our pleasure. What would we need to do over the next two weeks to strengthen the paper enough to raise your score?

---

### Official Review · Reviewer_dE5s · 2024-07-05

**Soundness:** 3
**Presentation:** 3
**Contribution:** 3
**Rating:** 6
**Confidence:** 4

**Summary:**

This paper presents a set of analyses on the dynamics with which internal language models’ mechanisms emerge and change during training. The four mechanisms studied are internal circuits that the model implements to carry out four simple tasks: indirect object identification (IOI), gendered pronoun, greater-than, and subject-verb agreement. These circuits have been identified by previous work and shown to be implemented by a set of specialized attention heads that display an interpretable behavior (e.g., induction heads, which attend to a previous occurrence of a substring that is repeated in the input).
The objects of the study are 7 models of the Pythia family, with sizes ranging from 70M to 12B parameters. The authors identify the components that implement each circuit using a technique called edge attribution patching with integrated gradients. In the first analysis, the authors show that the ability of the models to carry out the four tasks emerges roughly at the same point during training (in terms of # of tokens seen by the model). At the same point during training, a subset of the models’ attention heads are observed to start implementing the specialized behaviors used to implement the tasks studied, leading to the conclusion that these specialized heads are responsible for the emergence of the model performance.
In a second analysis, the authors study the dynamics which is how different sets of heads in Pythia 160M display the specialized behaviors relative to the four tasks considered. One observation is that the functional behavior of some heads decreases over time. This is surprising as this decrease is not reflected by the model’s performance. Copy suppression might be a possible explanation for this phenomenon.
Additionally, the paper includes an analysis of the stability of the algorithm implemented by the Pythia models on IOI, suggesting that the algorithm does not undergo significant changes after emerging during training.
Finally, the paper analyzes the changes in the actual components constituting the four circuits in Pythia 70M-2.8B.

**Strengths:**

- The analysis of circuit dynamics during language model pre-training is novel.
- The results about the emergence (and disappearance) of specialized heads, and about the models’ algorithmic stability during training are informative and are likely to be appreciated by the (mechanistic) interpretability community.

**Weaknesses:**

- Parts of the paper might benefit from additional details and clarification (see questions 1.1-1.3).
- The conclusions to be drawn from some of the results are not completely clear (see question 2).

**Questions:**

1. In Section 4.2, the authors measure, through path patching, the effect of a node in the computational graph in Figure 4A, on the model’s accuracy when performing IOI.
    1. As a reader, I would appreciate some additional details about the experimental procedure here: what are the exact heads that are being ablated? How are they selected for each model (e.g., the 5 heads with the highest CPSA score)?
     1. It is not clear to me why this effect is termed “direct.” As illustrated in Appendix B, path patching involves intervening on a node H (e.g., S-Inhibition heads), measuring the change produced by the intervention on a second node R that depends on H (e.g., NMH), and finally how the change in R an outcome variable Y (in this case, the model’s output). Through this procedure, what is being measured is the *indirect* effect of H on Y that is mediated by R.
     1. “For each step, our metric measures this direct effect, divided by the sum of the direct effects of ablating each edge with the same endpoint” (lines 254-255). I am confused about the denominator in this operation: what are the effects that are being summed here? Also, do the results look similar when unnormalized (i.e., when measuring the absolute effect)?


1. I am not sure whether there’s any clear conclusion that we can draw from the results presented in Section 5. It is true that the EWMA-JS for Pythia 70M seems to have a higher variance, but besides this, I struggle to see a clear trend in the measurements for the other models.
1. If you quantified the cumulative score for each type of head (e.g., induction score) over time for the top k heads, would it be constant after the point during training at which the model learns the task? This would confirm your hypothesis for which specialized heads are replaced by other heads as they lose their functional behavior during training (although my guess would be that such cumulative scores might be decreasing over time).

**Limitations:**

Limitations seem to be adequately discussed.

---

> ### Author Rebuttal · Authors · 2024-08-07
>
> Thanks for your review and helpful suggestions! We respond to your questions (which contain your stated weaknesses), below.
> > 1.1 As a reader, I would appreciate some additional details about the experimental procedure here: what are the exact heads that are being ablated? How are they selected for each model (e.g., the 5 heads with the highest CPSA score)?
>
> Here are additional details about our algorithmic consistency experiment, which we will add to the appendix. At each checkpoint, heads were selected for ablation as follows:
>
> 1. We ablated all heads upstream of a target or set of targets (e.g., the final logits or the set of name-mover heads) one at a time via path patching to determine their effect on the logit difference metric (either directly or through an intermediate node); this mirrors Wang et al. (2023).
>
> 2. We tested each of these heads for their target function (e.g., copy score for name-mover heads). We tested the set of heads that both A. had a component function score over a specific threshold (10%) and B. had a negative effect on logit difference when patched. For S2-inhibition heads, we tested whether ablating positional signal while keeping the token signal A. reduced logit difference, B. reduced NMH attention to the IO, and C. increased NMH attention to S1. See appendix D for the component tests for induction heads and duplicate token heads. All of these tests also come from Wang et al. (2023)
>
> > 1.2 It is not clear to me why this effect is termed “direct.” As illustrated in Appendix B, path patching involves intervening on a node H (e.g., S-Inhibition heads), measuring the change produced by the intervention on a second node R that depends on H (e.g., NMH), and finally how the change in R an outcome variable Y (in this case, the model’s output). Through this procedure, what is being measured is the indirect effect of H on Y that is mediated by R.
>
> You are right to question our use of the term direct in this paragraph. Upon re-reading this, we see that we inadvertently overloaded the term and should not have used “direct” here. The procedure is as you described.
>
> > 1.3 “For each step, our metric measures this direct effect, divided by the sum of the direct effects of ablating each edge with the same endpoint” (lines 254-255). I am confused about the denominator in this operation: what are the effects that are being summed here? Also, do the results look similar when unnormalized (i.e., when measuring the absolute effect)?
>
> Taking the notation you used in the previous question, if we take the numerator as the effect of ablating heads H (e.g., all S-Inhibition heads) on Y through intermediate heads R (e.g. NHMs), the denominator is the effect on Y of ablating heads G (all heads upstream of R, and includes H as a subset) through intermediate heads R.
>
> > 2. I am not sure whether there’s any clear conclusion that we can draw from the results presented in Section 5. It is true that the EWMA-JS for Pythia 70M seems to have a higher variance, but besides this, I struggle to see a clear trend in the measurements for the other models.
>
> We apologize for the lack of clarity here and agree that this section might have been better combined with Section 4. We aimed to quantify the shifting set of nodes that comprised each task circuit, and to illustrate that these shifts are not model-dependent. In most cases, shifts are gradual—Jaccard similarity is often between 0.6 and 0.9—but nevertheless real. We have added plots in our response PDF (Figures 1 and 2) that illustrate similarity to the final circuit. These show gradual changes result in a significant shift between the circuits at the beginning and end of training, despite our algorithmic consistency results.
>
> > 3. If you quantified the cumulative score for each type of head (e.g., induction score) over time for the top k heads, would it be constant after the point during training at which the model learns the task? This would confirm your hypothesis for which specialized heads are replaced by other heads as they lose their functional behavior during training (although my guess would be that such cumulative scores might be decreasing over time).
>
> This is a good suggestion, and one that we incorporated into Figure 3 of our response PDF. The figure now shows the sum of head effects across all in-circuit heads at a given checkpoint. Some trends remain the same as previously: the timestep at which components emerge is still tightly coupled to the emergence of task behavior. However, the degree to which the sum is constant over time varies across components; we believe that this reflects not only variation in the total component effects, but how much sense it makes to sum each component’s score.

---

> > ### Comment · Reviewer_dE5s · 2024-08-12
> >
> > Thank you for your thorough response. I appreciate the details provided and the additional analyses.
> >
> > > However, the degree to which the sum is constant over time varies across components; we believe that this reflects not only variation in the total component effects, but how much sense it makes to sum each component’s score.
> >
> > What can we conclude from these observations? Do you believe that copy suppression is the explanation for the decrease in the cumulative functional score not followed by a decrease in the model's performance on the tasks (Fig. 1), or are there other possible explanations for this phenomenon?

---

> > > ### Author Response · Authors · 2024-08-12
> > >
> > > This seems like a potentially interesting question, but we’re not sure we’ve understood it correctly! Just to clarify: the copy suppression and name mover heads are (in Pytha models) the two families of components with direct effects on IOI task performance. We see in Rebuttal Fig. 1 that while summed copy suppression scores exhibit a slight downward trend toward the end of training, and Pythia-160m’s summed name mover head score decreases near the end of training, IOI performance does not decrease (Main paper Fig. 1).
> > >
> > > Why might this occur? While in GPT-2 small (studied by Wang et al. (2023)), the name-mover heads were the most important part of the IOI circuit, in Pythia models, the copy suppression heads are much more significant. As only small changes occurred in the more important component in the circuit (the CS heads) in Pythia, model behavior doesn’t change very much.
> > >
> > > Perhaps more importantly, while our e.g. copy suppression head metric verifies that a head does copy suppression generally, this metric does not try to quantify the strength of copy suppression on the given task. It's plausible that these heads could perform differently on different data distributions. Note that this is by design—our metrics measure behaviors regardless of task relevance. If we want to measure task relevance, it might be better to combine head-level importance scores with the component metrics to ask the question, e.g. “What is the total effect of copy suppression heads in my circuit, weighted by how much they behave like copy suppression heads?” We already ensure some task relevance by summing scores only on in-circuit nodes, but this new weighted metric might better capture the quantity you’re discussing, and would be valuable.
> > >
> > > The big conclusion for us was how much clearer the emergence points of these heads became; it’s now much more visible that their emergence directly precedes task performance. In our earlier plots, we showed the emergence of one individual successor / induction / etc. head per model, but this was flawed; in the early stages of component emergence, several competing heads often developed at once. As a result, it was hard to see component emergence when plotting just one head. By considering the emergence across all heads in the circuit, we were able to see the emergence of these heads overall, even before any particular head emerged.

---

> > > > ### Comment · Reviewer_dE5s · 2024-08-13
> > > >
> > > > Thank you for your response. I will be raising my score to reflect the author's clarifications (which should be included in the paper) and the additional analyses.

---

### Author Rebuttal · Authors · 2024-08-07

We would like to thank all the reviewers for their thoughtful responses! We are glad to hear that reviewers feel that our paper:
- Presents novel and interesting insights (dE5s, v6cM, wt3i)
- Extends beyond prior work by studying models across scales (865w, wt3i, v6cM)
- Is valuable to the mechanistic interpretability community (dE5s, 865w)

However, the reviewers also shared some critiques in common, which we would like to address here:
- **This work studies only a small number of simple tasks (865w, wt3i, v6cM)**: This is true, and due in part to our desire to study models of a varying sizes: small models often fail on complex tasks, so studying their behavior on large tasks is not insightful. Moreover, there are few complex tasks for which circuits have been found. Finally, because we already study models across two axes (scale and time), a third axis (tasks) would have been too computationally expensive. However, we agree that this aspect is important, and hope to investigate it in future work.
- **Potential flaws with our analysis of model components in section 3.2** (dE5s, 865w, v6cM): Reviewer dE5s notes that our analysis is limited by the fact that we consider only individual components, rather than the sum of their effects, while reviewer 865w notes that we analyze components without regard to whether the are in the circuit. These criticisms have merit, so we re-analyzed our data via the following procedure: Instead of plotting individual heads, we sum scores across heads for each checkpoint, and normalize, dividing by the maximum sum across checkpoints. We set the score of any head not in the circuit to 0. We plot this in Figure 1, using the same x-axis as our behavioral plots as requested by reviewer v6cM. We find that the point at which components develop in this plot more closely tracks the emergence of task behavior than it did in our previous plot. However, trends in the sum across epochs are variable across tasks: while most tasks’ sums rise over training, induction score changes wildly. We attribute this to not only actual variation in the number of heads acting as a component, but also measurement issues: by taking the sum across all in-circuit heads, we might include many heads that have a low but non-zero component score, obscuring trends in heads that actually perform that component ability.
- **Requested clarifications about our algorithmic stability analysis in section 4.2 (dE5s, v6cM)**: Reviewers dE5s and v6cM requested details about our algorithmic stability experiments in Section 4.2; specifically, they asked how we selected heads for ablation, and whether these were part of the circuit (and thus relevant to model behavior). We would like to clarify that we selected heads based on two criteria: first, we selected heads based on their component scores as in Section 3.2. Second, we selected heads that had a large effect when targeted using the path-patching causal intervention; this path-patching effect is roughly the quantity estimated by EAP-IG when we find circuits. Thus, we actually use a more accurate method of finding important components in Section 4.2. For more details on this procedure, see our responses to dE5s and v6cM.
- **Questions about circuit similarity in section 5 (dE5s, wt3i, v6cM)**: Reviewer dE5s asked about the interpretation of Figure 5, and reviewer wt3i remarked that it was difficult to see if circuits changed significantly versus only the previous checkpoint, as we measured weighted similarity with all previous checkpoints. Reviewer v6cM also suggested measuring the Jaccard similarity of circuit components weighted by importance.
In our PDF response, we provide two plots: one with the given checkpoint’s circuit’s unweighted node similarity to the final circuit (Figure 2), and one with its weighted edge similarity to the final circuit; note that we cannot perform weighted node similarity, as EAP-IG only yields edge weights (Figure 3). Both plots indicate that while inter-checkpoint similarity to the final circuit varies, circuits eventually become more and more similar to the final checkpoint. We also add that our implementation of EWMA does result in scores that are mostly influenced by nearby checkpoints, as the weighting of JS to distant checkpoints drops off rapidly. As for the intention of these plots: our objective was to quantify the level of change that occurs in the constituents of these circuits even as performance largely remains stable beyond a certain point.

---

### Decision · Program_Chairs · 2024-09-25

**Decision:**

Accept (poster)

**Comment:**

This work analyzes the stability of circuits in language models during pretraining and across different model sizes using the set of Pythia models, showing a significant consistency in attention heads and circuits across training and scale. In particular, the work shows that specialized attention heads responsible for tasks like indirect object identification (IOI), gendered pronouns, greater-than, and subject-verb agreement emerge consistently during training and maintain stable learning across model sizes after a similar number of tokens during training. Some reviewers thought surprising that some heads show a decline in functional behavior over time, potentially due to copy suppression.

The authors clarified some points and performed additional experiments during the rebuttal, which I found to be both thoughtful and insightful. The reviewers were quite positive about the work. While some of the phenomena observed in this work need validation by others, I suggest acceptance.

If accepted the authors need to include the clarifications in the paper, it will definitively help future readers.